# A Novel Benzopyrane Derivative Targeting Cancer Cell Metabolic and Survival Pathways

**DOI:** 10.3390/cancers13112840

**Published:** 2021-06-07

**Authors:** Dana M. Zaher, Wafaa S. Ramadan, Raafat El-Awady, Hany A. Omar, Fatema Hersi, Vunnam Srinivasulu, Ibrahim Y. Hachim, Farah I. Al-Marzooq, Cijo G. Vazhappilly, Salim Merali, Carmen Merali, Nelson C. Soares, Paul Schilf, Saleh M. Ibrahim, Taleb H. Al-Tel

**Affiliations:** 1Sharjah Institute for Medical Researches, University of Sharjah, Sharjah 27272, United Arab Emirates; U17105878@sharjah.ac.ae (D.M.Z.); U17105877@sharjah.ac.ae (W.S.R.); relawady@sharjah.ac.ae (R.E.-A.); hanyomar@sharjah.ac.ae (H.A.O.); fhersi@sharjah.ac.ae (F.H.); svunnam@sharjah.ac.ae (V.S.); ihachim@sharjah.ac.ae (I.Y.H.); f.almarzooq@uaeu.ac.ae (F.I.A.-M.); cijo.vazhappilly@aurak.ac.ae (C.G.V.); nsoares@sharjah.ac.ae (N.C.S.); saleh.ibrahim@uksh.de (S.M.I.); 2College of Medicine, University of Sharjah, Sharjah 27272, United Arab Emirates; 3College of Pharmacy, University of Sharjah, Sharjah 27272, United Arab Emirates; 4Faculty of Pharmacy, Beni-Suef University, Beni-Suef 62511, Egypt; 5Department of Medical Microbiology and Immunology, College of Medicine and Health Sciences, United Arab Emirates University, Al Ain 15551, United Arab Emirates; 6School of Arts and Sciences, American University of Ras Al Khaimah, P.O. Box 10021, Ras Al Khaimah 10021, United Arab Emirates; 7School of Pharmacy, Temple University, 3307 N Broad Street, Room 552, Philadelphia, PA 19140, USA; smerali@temple.edu (S.M.); clmerali@temple.edu (C.M.); 8Lübeck Institute of Experimental Dermatology, University of Lübeck, Ratzeburger Allee 160, 23538 Lübeck, Germany; Paul.Schilf@uksh.de

**Keywords:** multitarget, apoptosis, DNA damage, cell cycle, thioredoxin reductase, glutathione reductase

## Abstract

**Simple Summary:**

This work aimed to illustrate the anticancer mechanism of action of a novel benzopyrane derivative as a potential anticancer lead compound. The anticancer activity of SIMR1281 against a panel of cancer cell lines was characterized. The effects of SIMR1281 on glutathione reductase (GSHR), thioredoxin reductases (TrxR), mitochondrial metabolism, DNA damage, cell cycle progression, and apoptosis induction were determined. SIMR1281 was evaluated in vivo for its safety and efficacy. SIMR1281 strongly inhibited GSHR while it moderately inhibited TrxR and modulated the mitochondrial function. It inhibited cell proliferation by inducing DNA damage, perturbations of the cell cycle, and inactivation of Ras/ERK and PI3K/Akt pathways, consequently leading to apoptosis. SIMR1281 significantly reduced tumor volume in a tumor xenograft model while maintaining a high safety. These findings support developing SIMR1281 in preclinical and clinical settings as it represents a potential lead compound for the treatment of cancer.

**Abstract:**

(1) Background: Today, the discovery of novel anticancer agents with multitarget effects and high safety margins represents a high challenge. Drug discovery efforts indicated that benzopyrane scaffolds possess a wide range of pharmacological activities. This spurs on building a skeletally diverse library of benzopyranes to identify an anticancer lead drug candidate. Here, we aim to characterize the anticancer effect of a novel benzopyrane derivative, aiming to develop a promising clinical anticancer candidate. (2) Methods: The anticancer effect of SIMR1281 against a panel of cancer cell lines was tested. In vitro assays were performed to determine the effect of SIMR1281 on GSHR, TrxR, mitochondrial metabolism, DNA damage, cell cycle progression, and the induction of apoptosis. Additionally, SIMR1281 was evaluated in vivo for its safety and in a xenograft mice model. (3) Results: SIMR1281 strongly inhibits GSHR while it moderately inhibits TrxR and modulates the mitochondrial metabolism. SIMR1281 inhibits the cell proliferation of various cancers. The antiproliferative activity of SIMR1281 was mediated through the induction of DNA damage, perturbations in the cell cycle, and the inactivation of Ras/ERK and PI3K/Akt pathways. Furthermore, SIMR1281 induced apoptosis and attenuated cell survival machinery. In addition, SIMR1281 reduced the tumor volume in a xenograft model while maintaining a high in vivo safety profile at a high dose. (4) Conclusions: Our findings demonstrate the anticancer multitarget effect of SIMR1281, including the dual inhibition of glutathione and thioredoxin reductases. These findings support the development of SIMR1281 in preclinical and clinical settings, as it represents a potential lead compound for the treatment of cancer.

## 1. Introduction

Most of the anticancer drugs currently in use are single-target drugs. This prototype is becoming less effective in managing complex, polygenic diseases, such as cancer [1,2]. To overcome these challenges, a combination of drugs is used as an alternative method. However, their remunerations are often associated with adverse effects, including drug–drug interaction, unexpected pharmacokinetics (PK) and safety profiles, and poor patient compliance [3,4]. Therefore, there have been pronounced efforts towards discovering "multitarget drugs" to overcome these challenges [5,6]. A growing body of evidence suggests that increased levels of thioredoxin reductase (TrxR) or glutathione reductase (GSHR) have been detected in multiple tumors, including colorectal, breast, and lung cancers [7]. Therefore, treatments centered on silencing the thioredoxin (Trx) and glutathione (GSH) pathways epitomize a novel methodology to treat cancer and to enhance the defense mechanism by the immune system [8].

Moreover, the dual inhibition of the redox–redox systems Trx and GSH has been shown to synergistically kill tumor cells in vitro and in vivo and reduce resistance to anticancer therapy [8]. It has also been reported that the inhibition of Trx and GSH may lead to reprogramming of the immune response, regulating the balance between the immune system and cancer, favoring the former, and permitting the elimination of cancer cells [8]. Besides, an essential property of an effective chemotherapy drug is its ability to cause DNA damage by inducing harmful DNA lesions, leading to apoptosis [9,10]. This mechanism was validated by the drug’s ability to enhance the phosphorylation of ataxia telangiectasia-mutated (ATM) or ATM-Rad3-related kinases (ATR) as well as the phosphorylation of Ser139 residues on specialized γ-H2AX histones [11]. Importantly, the downregulation of cancer survival proteins remains a challenge in developing new cancer treatment approaches [12,13]. Several reports indicated that targeting pro-apoptotic and anti-apoptotic signaling pathways in cancers along with multiple key enzymes required for cancer survival and spread, such as TrxR, GSHR, inositol-3-phosphate synthase, phosphatidylinositol-3-kinase (PI3K)/protein kinase B (Akt) and a Ras/extracellular kinase regulated by signaling (ERK), represent a unique approach that disrupts the evolutionary dynamics of the cancer ecosystem [14]. 

In this study, we report the discovery of a first-in-class small molecule SIMR1281, which modifies the function of many key biomolecules needed for cancer survival, proliferation, replication, and spread. SIMR1281 has been shown to promote DNA damage and subsequent mitotic aberrations, preventing cancer spread in vitro and in vivo. SIMR1281 might serve as a clinical anticancer candidate with a unique pleiotropic mechanism of action, and here, we report our findings.

## 2. Materials and Methods

### 2.1. Cell Culture

Human breast adenocarcinoma cell lines (MCF7 [RRID: CVCL_0031], SKBR3 [RRID:CVCL_0033], BT549 [RRID:CVCL_1092], and MDA-MB231 [RRID:CVCL_0062]), human non-small cell lung cancer cell line (A549) [RRID:CVCL_0023], human colorectal carcinoma cell line (HCT116) [RRID:CVCL_0291], human histiocytic lymphoma cell line (U937) [RRID:CVCL_0007], human acute T cell leukemia cell line (Jurkat) [RRID:CVCL_0367], Abelson murine leukemia cell line (Raw 246.7) [RRID:CVCL_0493] and human normal fibroblast cell line (F-180) were obtained from the Radiobiology and Experimental Radio Oncology lab, University Cancer Center, Hamburg University, Hamburg, Germany. Doxorubicin-resistant MCF7 and A549 cell lines were generated in our lab. MCF7, SKBR3, MDA-MB231, and A549 cells were maintained in RPMI-1640 medium (Sigma-Aldrich-St. Louis, MO, USA). Media were supplied with 10% fetal bovine serum (Sigma-Aldrich-St. Louis, MO, USA) and 1% penicillin/streptomycin (Sigma-Aldrich-Louis, MO, USA). HCT116, U937, Jurkat, Raw 246.7, and F-180 were cultured in DMEM medium (Sigma-Aldrich-St. Louis, MO, USA), which is supplied with 10% fetal bovine serum (Sigma-Aldrich-St. Louis, MO, USA) and 1% penicillin/streptomycin (Sigma-Aldrich-St. Louis, MO, USA). All incubations were performed at 37 °C in a humidified atmosphere of 5% CO_2_.

### 2.2. Cell Viability Analysis

Sulforhodamine B (SRB) and 3-(4,5-dimethylthiazol-2-yl)-2,5-diphenyltetrazolium (MTT) cell viability assays were conducted in a panel of cancer cell lines including MCF7, SKBR3, BT549, MDA-MB231, A549, HCT116, U937, Jurkat, and Raw 246.7 murine cells as well as normal fibroblast F-180 cells to determine the half-maximal inhibitory concentration (IC_50_) of SIMR1281. Briefly, cells were seeded at a density of 1 × 10^4^ per well in 96-well plates overnight and were treated for 48 h with several concentrations of SIMR1281. The wells which were devoid of any treatment served as control. After the treatment time, for the SRB assay, the plates were incubated with 50–80% trichloroacetic acid (TCA) at 4 °C for 1 h. Cells were then washed, dried, and exposed to 1% acetic acid for a short period. Plate centrifugation was performed for U937, Jurkat, and Raw 246.7 murine cells to settle down the cells. Then, cells were incubated for 10 min with 200 µL of 10 mM Tris-base solution. For the MTT assay, after treatment, the media was replaced by fresh media containing 0.5 mg/mL of MTT—Thiazolyl Blue Tetrazolium Bromide reagent (Sigma-Aldrich, St. Louis, MO, USA, Cat# M2128). Finally, after 2 h incubation at 37 °C, DMSO (Sigma-Aldrich, St. Louis, MO, USA, Cat# 67-67-5) was added, and the formed formazan crystals were solubilized. The ODs were measured at 492 or 570 nm for SRB and MTT, respectively, using the Varioskan Flash (Thermo Fisher Scientific, Waltham, MA, USA) microplate spectrophotometer. 

### 2.3. Western Blot Analysis

MCF7, SKBR3, and HCT116 total cell lysates were collected after 24 h treatment with SIMR1281 at IC_50_ and twice the IC_50_ (2 × IC_50_) doses, and 1× Laemmli buffer was added. Then, the total protein concentration was determined, and 15–30 µg of protein lysates were loaded equally on 12% SDS-PAGE gel to examine various protein expression levels. The proteins were transferred to the membrane (0.45 µM nitrocellulose, Cat# 1620115, Bio-rad, California, CA, USA). After blocking, the membrane was incubated with various primary antibodies, which were purchased from Cell Signaling Technology (Danvers, Massachusetts, MA, USA): p-H2Ax Ser139 (Cat# 9718, Rabbit IgG, Clone# 20E3, RRID:AB_2118009), H2Ax (Cat# 2595, Rabbit, RRID:AB_10694556), p-ATM Ser1981 (Cat# 5883, Rabbit IgG, Clone# D6H9, RRID:AB_10835213), ATM (Cat# 2873, Rabbit IgG, Clone# D2E2, RRID:AB_2062659), p-ATR Ser428 (Cat#2853, Rabbit, RRID:AB_2290281), ATR (Cat# 2790, Rabbit, RRID:AB_2227860), c-Myc (Cat# 5605, Rabbit IgG, Clone# D84C12, RRIP:AB_1903938), p53 (Cat# 2524, Mouse IgG1, Clone# 1C12, RRID:AB_2256294), p21 (Cat# 2947, Rabbit IgG, Clone# 12D1, RRID:AB_823586), Bax (Cat# 5023, Rabbit IgG, Clone# D2E11, RRID:AB_10557411), GAPDH (Cat# 2118, Rabbit IgG, Clone# 14C10, RRID:AB_561053), caspase-3 (Cat# 9665, Rabbit IgG, Clone# 8G10), caspase-9 (Cat# 9508, Mouse IgG1, Clone# C9, RRID:AB_2068620), p-Chk1 Ser345 (Cat# 2348, Rabbit IgG, Clone# 133D3, RRID:AB_331212), Chk1 (Cat# 2360, Mouse IgG1, Clone# 2G1D5, RRID:AB_2080320), p-Chk2 Thr68 (Cat# 2197, Rabbit IgG, Clone# C13C1, RRID:AB_2080501), Chk2 (Cat# 2662, Rabbit, RRID:AB_2080793), Ras (Cat# 3965, Rabbit, RRID:AB_2180216), p-ERK1/2 Thr202/Tyr204 (Cat# 4377, Rabbit IgG, Clone# 197G2, RRID:AB_331775), ERK1/2 (Cat# 9102, Rabbit, RRID:AB_330744), p-Akt Ser473 (Cat# 4060, Rabbit IgG, Clone# D9E, RRID:AB_2315049), Akt (Cat# 9272, Rabbit, RRID:AB_329827) and Cyclin B1 (Cat# 12231, Rabbit IgG, Clone# D5C10, RRID:AB_2783553). Cyclin A (Cat#sc-751, Rabbit, Clone# H-432, RRID:AB_631329) was purchased from Santa Cruz Biotechnology (Dallas, TX, USA). β-actin (Cat# A5441, Mouse, RRID:AB_476744) was purchased from Sigma-Aldrich (St. Louis, MO,USA). The used antibodies dilution was 1:1000. After overnight incubation, membranes were blocked with 5% non-fat milk solution for 1 h. Then, membranes were re-probed for 1 h with mouse/rabbit secondary antibodies from cell signaling technology according to the primary antibody source (Anti-mouse IgG, HRP-linked, Cat# 7076, Horse, RRID:AB_330924 and Anti-rabbit IgG, HRP-linked, Cat# 7074, Goat, RRID:AB_2099233) (1:2000). Finally, membranes were developed using enhanced chemiluminescence (ECL) method by using a Chemidoc MP (Bio-rad, California, CA, USA).

### 2.4. Annexin V Staining Assay

The apoptotic level was assessed using the FITC Annexin V Apoptosis Detection Kit (BD Pharmingen, San Diego, CA, USA, Cat# 556547) as per the supplier’s protocol. MCF7 and F-180 cells were seeded at a density of 3 × 10^6^ cells per T-75 cm2 flask and incubated overnight to attain 70% confluency. Then, cells were subjected to treatment with IC_50_ doses of SIMR1281 along with doxorubicin for 12 and 24 h. Cells were then scraped off and washed twice with ice-cold PBS. Then, 2 × 10^6^ cells were counted and re-suspended in 1 mL of binding buffer (1×). After that, 100 µL of cell suspension was incubated for 15 min with 5 µL of FITC Annexin V and PI in the dark. Finally, 400 µL of 1× binding buffer was added to cells and analyzed using a flow cytometer (Accuri C6, BD Pharmingen, San Diego, CA, USA). 

### 2.5. Cell Cycle Analysis

Cell cycle arrest of SIMR1281 was analyzed by an established protocol with minor modifications [15]. In brief, 3 × 10^6^ cells of MCF7, HCT-116, and F-180 cells were seeded and incubated overnight in a T75 cm2 flask before being treated with the IC_50_ dose of SIMR1281 for 16, 24, 48, and 72 h. Cells that were devoid of any treatments for each group served as control. After the respective treatment hours, cells were washed twice with PBS and fixed with ice-cold ethanol (70%) overnight. Cells were then washed twice with cold PBS, and 1 × 10^6^ cells were then counted and incubated with 1 mL of RNAase (100 µg/mL) for 30 min at 37 °C. The cell pellet was further added with 200 µL of propidium iodide (50 µg/mL) and immediately analyzed by a flow cytometer (Accuri C6, BD Pharmingen, San Diego, CA, USA). 

### 2.6. Immunofluorescence Assay

An immunofluorescence assay was employed to detect the effect of SIMR1281 on tubulin polymerization [16]. A549 cells (5 × 10^4^/well) were plated overnight on coverslips on a six-well plate and treated with the IC_50_ concentration of SIMR1281, taxol, and colchicine for 24 h. After treatment, cells were rinsed twice with PBS, fixed with 3.7% paraformaldehyde, and permeabilized with 0.1% Triton X-100. Cells were then blocked with 1% BSA in PBS for 1 h before further incubation with an anti-β-tubulin monoclonal antibody (Cell signaling Cat# 86298, Mouse IgG2b, Clone# D3U1W) overnight at 4 °C (Cell Signaling, San Francisco, CA, USA). Then, cells were incubated with anti-mouse Alexa Fluor^®^ 488 secondary antibody (Abcam, Cambridge, MA, USA, Cat# 150117, goat), after being washed with PBS for 1 h in the dark. Finally, the cellular microtubules were detected with a confocal laser-scanning microscope (Nikon Eclipse Ti Microscopy, MO, USA).

### 2.7. In Vitro Tubulin Polymerization Assay

Tubulin polymerization kinetics was assessed in vitro using a tubulin polymerization assay kit (Cytoskeleton #BK011P, Denver, CO, USA). In brief, 2 mg/mL porcine tubulin was dissolved in buffer 1 (80 mM PIPES, 2 mM MgCl2, 0.5 mM EGTA, pH 6.9, 10 µM fluorescent reporter, 1 mM GTP, 15% glycerol) to reach a concentration of 10 mg/mL. The tubulin solution was then transferred to a pre-warmed 96-well plate containing the IC_50_ dose of SIMR1281 along with 0.5 µM CaCl2 and 3 µM paclitaxel. Tubulin polymerization was monitored as a fluorescence signal at 37 °C for 60 min at an excitation wavelength of 360 nm and an emission wavelength of 450 nm. Reading was carried out using the Multi-skan go (Thermo Fisher Scientific, Waltham, MA, USA).

### 2.8. DARTS Assay

The Drug Affinity Response Target Stability assay (DARTS) was used to assess the association of SIMR1281. HEK293T cells were analyzed with M-PER (Thermo Fisher Scientific, Waltham, MA, USA) supplemented with proteases and phosphatase inhibitors (lysis solution). A total of 4.7 mg/mL protein was added, TNC solution (10 mM Tris, 140 mM NaCl, 5 mM calcium chloride), SIMR1281 was diluted in DMSO at various concentrations (0.01 μM, 0.1 μM, 1 μM), and incubated for 1 h at room temperature. Trypsin was added (Promega, North Carolina, NC, USA and incubated for 3 h at 37 °C, then M-PER lysis solution was added, and samples were concentrated using Microcon 10K centrifugal filters. The proteins were identified by liquid gel mass spectroscopy (GeLCMS), as described previously [17,18]. Raw data were analyzed using MaxQuant version 1.6.10.43 (Max Planck Institute of Biochemistry, Planegg, Germany), and MS/MS spectra were examined using the Andromeda search engine against the Uniprot Human Protein Database. The maximum allowable mass tolerance was set to 20 ppm for the precursor and then set to 6 ppm in the main search and 0.5 Da for fragment ions. Enzyme specificity to trypsin was adjusted with a maximum of two missed divisions. Carbamidomethyl cysteine was designated as the static modification, and methionine oxidation and protein N-acetylcysteine were selected as covariate modifications. The minimum peptide length was set to 6 amino acids. Label-free protein quantification was carried out using a label-free quantification (LFQ) algorithm previously described in the MaxQuant software (Max Planck Institute of Biochemistry, Planegg, Germany) with a 2-min window for matching between runs and a protein false detection rate of 1% maximum and 1% protein. The protein intensity values were normalized using the LFQ algorithm available through the MaxQuant software and used to further assess the differential abundances of the identified proteins. Bioinformatics analyses of data were performed with Prism version 8 (GraphPad Software, Inc., La Jolla, CA, USA).

### 2.9. Mitochondrial Membrane Potential

Experiments were performed using the Hepa1-6 cell line (# CRL-1830, ATCC). Cells were cultured in DMEM media with 10% fetal bovine serum (FBS), 1 g/L glucose, 2 mM glutamine, 100 units/mL penicillin, and 100 mcg/mL streptomycin and 30 μM oleic acid. After seeding, 30,000 cells per well in 96 well plates, and cells were incubated at 37 °C with 5% CO2 overnight to let the cells attach to the culture surface. Mitochondrial membrane potentials (μM) were assessed with a Tetramethylrhodamine (TMRE) fluorophore (Thermo Fisher Scientific, Waltham, MA, USA, # T669). Cells were treated with SIMR1281 for 24 h. Then, TMRE was added to the cells at 200 nM and incubated for 45 min at 37 °C. In separate control wells, incubation with FCCP (10 µM) for 30 min before the addition of TMRE was used as a positive control. Cells were washed once with PBS, and a fresh culture medium was added before measurement. TMRE fluorescence was measured at Ex/Em = 545/580 nm. Signals in the vehicle-treated and blank-well controls were used to normalize the results.

### 2.10. Analysis of ATP Content Assay

After treating the cells with SIMR1281 at the specified concentrations for 24 h, the cellular ATP content was assessed with the CellTiter-Glo assay (Promega, North Carolina, NC, USA, # G757). A plate reader spectrophotometer was used to determine ATP-dependent luminescence. The luminescence measured in vehicle-treated wells and well-treated controls with 10 μM oligomycin were used to normalize the results.

### 2.11. Analysis of Cellular Redox Potential

After the treatment of cells with 10 μM SIMR1281 or controls (10 μM oligomycin, 10 μM FCCP, or 0.1% DMSO) for 20 h, the culture media was supplemented with 1/10 volume of AlamarBlue reagent (resazurin, Bio-rad, California, CA, USA, # BUF012A) and incubated for another 4 h. The reduction of resazurin was measured as the fluorescence intensity at Ex/Em = 550/590 nm. Fluorescence signals in vehicle-treated wells and blank-well controls were used to normalize the results.

### 2.12. Thioredoxin Reductase and Glutathione Reductase Inhibition Assays 

Thioredoxin reductase activity was determined using a thioredoxin reductase assay kit (Abcam, Cambridge, MA, USA, Cat # ab83463). Briefly, HCT116 cells were treated with 0.2, 2, 4, 8, 10 μM of SIMR1281 for 48 h. Cells were lysed, and protein quantification was performed. After adjusting protein concentrations, two aliquots from the same sample were tested either with TrxR inhibitor or without it. Ten microliters of TrxR inhibitor was added to one set of the sample, and 10 μL of assay buffer was added to the other set. After adding the reaction mixture, OD was measured directly at 412 nm using a Varioskan Flash Multimode Spectral Scan Reader (Thermo Fisher Scientific, Waltham, MA, USA). OD was measured again after incubation for 20 min. Data analysis was performed according to the manufacturer’s instructions. Reduced glutathione (GSH) was quantified using a GSH/GSSG ratio detection assay kit (Abcam, # ab138881) as per the manufacturer’s instructions. After treating HCT116 cells with 0.4, 0.8, 1.5, 3, and 6 μM of SIMR1281 for 48 h, cells were harvested and lysed using 0.5% NP-40 lysis buffer. The TCA/NaHCO3 protein deproteinization step was performed, and the GSH assay mixture was applied to the samples. GSH was measured directly using a GSH standard; Fluorescence was monitored at Ex/Em = 490/520 nm using Multiskan go (Thermo Fisher Scientific, Waltham, MA, USA). 

#### Ethics Statement

The number of animals, study design, and treatment protocol were reviewed and approved by the University of Sharjah Animal Care and Use Committee (Approval# ACUC-18-12-17-01). Furthermore, all procedures in this protocol were firmly performed according to the Guide for the Care and Use of Laboratory Animals published by the US National Institute of Health (NIH publication No. 80-23, revised 1996). 

### 2.13. Mice Models

A total of 21 male Adult SJL/J mice and 12 female NU/J mice weighing 18–25 gm were purchased from the Jackson Laboratory (Bar Harbor, ME) and housed in Sharjah Institute for Medical Research, University of Sharjah, at a constant temperature (25 ± 2 °C), humidity (60 ± 10%) and a 12/12 h light/dark cycle. Mice were provided with standard chow and water ad libitum. Animals were acclimatized for 7 days before starting the experimental work. To achieve blindness, different people conducted the experiments and the analysis. 

### 2.14. In Vivo Safety Studies 

The study duration was divided into single and multiple-dose phases to identify the highest dose of SIMR1281 that does not cause undesirable side effects or overt toxicity. In the single-dose toxicity study, we started with the sighting study, which allows the selection of the appropriate starting dose for the main study [19,20]. Thus, SIMR1281 (5, 10, 20, 40, 80, and 160 mg/kg) or vehicle (0.5% DMSO in PBS) was administered through i.p injection to single adult SJL/J mice. A period of at least 24 h was allowed between the testing of each animal. When the selected dose did not cause acute death or a significant change in body weight within 48 h of drug administration, we started the main study where two more mice were introduced to that group and were observed daily for 16 days. A total of three animals were used for each dose level investigated. The doses in between the range that did not cause any unacceptable side effects or overt toxicity in mice with a single-dose were used for the multiple-dose toxicity study. Animals were divided into 3 groups randomly, which received daily doses of vehicle or test substance SIMR1281 (25, 50 mg/kg) through i.p. injection for 14 days. Animals were then subjected to follow-up observations for 12 days post-treatment to detect any possible reversibility, persistence, or delayed occurrence of toxic effects. 

### 2.15. Tumor Xenograft Mice Model Implantation 

HCT116 cells were suspended at a density of 2 × 10^6^/50 μL of FBS-free media. Matrigel was added to the cell suspension in 1:1 ratio. Female nude mice which were 8–10 weeks old were subcutaneously inoculated with 100 μL of cell suspension. The skin was tented up, and HCT116 cells were implanted under the skin. Mice were divided into 2 groups; SIMR1281 group (*n* = 7) received daily 50 mg/kg of the compound through i.p injection for 30 days while the control group (*n* = 5) received the vehicle (40% PEG400 + 4%DMSO + PBS). Tumor length (L) and width (W) were measured twice weekly using calipers, and tumor volume (V) was calculated as (V = (L × W2)/2). Mice weights were monitored daily. After 30 days of treatment, mice were sacrificed, and blood was collected for a complete blood count test and clinical chemistry parameters analysis using the Pchem-1 Chemistry Auto analyzer (Serial No. A-1511) (Via Luigi Einaudi, Italy) and Hematology Analyzer DxH 520 (Beckman Colter, California, CA,USA). In addition, tumors and selected organs were collected for subsequent histopathological examination. 

### 2.16. Hematoxylin and Eosin Staining and Immunohistochemistry

For microscopic histological evaluation, the collected organs (liver, kidney, spleen, heart) and the tumors were subjected to fixation step in 10% neutral-buffered formalin for 24 h. Paraffin blocks were prepared for immunohistochemistry as detailed before [21]. Histological scoring as follows: Nil = 0; mild = 1; moderate = 2 and severe = 3, was performed by an investigator who was blinded to the experimental groups. For immunohistochemical staining, tumor tissue sections were stained with Anti-Ki67 (Abcam, Cambridge, MA, USA, #ab16667). For microscopic analysis, three representative fields per section per sample were randomly selected, and the mean number of Ki67 expressing cells was calculated. 

### 2.17. Data and Statistical Analysis

Results are presented as mean ± SEM. Curve fittings, data, and statistical analysis were carried out by the GraphPad Prism program (Prism 5, San Diego, CA, USA). Comparisons of two groups were made using the Student’s *t*-test. The comparison of multiple groups was performed by one-way ANOVA with Dunnett’s post hoc test unless otherwise stated. Non-linear regression was fitted to the data points using GraphPad.

## 3. Results

### 3.1. SIMR1281 Antiproliferative Effect on a Panel of Cancer Cell Lines 

The recent decade has witnessed an upsurge in the employment of diversity-oriented synthesis strategies for the de novo construction of first-in-class small molecules needed for phenotypic screening campaigns [22,23,24]. Among the structural options, the benzopyrane scaffold is a privileged architecture representing the core structure of a wide range of biologically appealing molecules [25]. We have recently reported a modular route that led to the access of a skeletally diverse library of benzopyranes [26,27]. Testing the potential anticancer activity of this pilot library in a panel of cancer cell lines yielded a unique chemotype, compound SIMR1281, which effectively suppressed cancer cell proliferation (Figure 1a, Appendix A). 

The developed compound library (Appendix A) was initially screened for potential antiproliferative activity against breast cancer cell lines (MCF7 and SKBR3) and colorectal cancer cell line (HCT116) at 10 μM. This study revealed that SIMR1281 compound possesses the most potent anticancer activity (Figure 1a, Appendix A). To determine whether SIMR1281 inhibits the proliferation of various cancers, it was screened against a panel of cancer cell lines, including cancer cells that had acquired drug resistance. The antiproliferative activity of SIMR1281 was tested at various concentrations on human breast adenocarcinoma cell lines (MCF7, SKBR3, BT549, and MDA-MB231), human non-small cell lung cancer cell line (A549), and human colorectal carcinoma cell line (HCT116) (Figure 1b). In addition, the anticancer activity of SIMR1281 was also tested against liquid cancer cell lines, including the human histiocytic lymphoma cell line (U937), human acute T cell leukemia cell line (Jurkat), and the Abelson murine leukemia cell line (Raw 246.7) (Figure 1c). The IC_50_ values of SIMR1281 range between 0.66 and 5.5 μM (Figure 1d). SIMR1281 indicates a high safety profile as determined from normal fibroblast (F-180) treatment, which exhibited a higher IC_50_ in F-180 cells (7.35 μM) when compared to cancer cells (Figure 1d,e). Another major impediment encountered in cancer chemotherapy is the development of resistance to medications. Therefore, we investigated whether SIMR1281 maintains its potency against MCF7 and A549 cancer cells that had acquired resistance to the conventional chemotherapeutic agent, doxorubicin (Dox). This study revealed that SIMR1281 is equally effective against these resistant cell lines (Figure 1f, Appendix A). These promising findings encouraged us to further study the molecular mechanism of action of SIMR1281.

### 3.2. SIMR1281 Modulates the Functions of Key Proteins and Mitochondria

To uncover the mechanism of action of SIMR1281, the DARTS proteomic strategy was performed to identify the cellular targets of SIMR1281 [28]. This assay revealed a strongly protected band at 150 kDa in extracts of SIMR1281-treated lysate (Appendix A). Excised gel bands corresponding to the protected proteins were further subjected to mass spectrometric analysis. The collected data were analyzed on an Orbitrap Elite mass spectrometer using Maxquant. MS data analysis of a library of proteins identified at least two peptides with Q-values = 0 and Andromeda scores between 6.5 and 323. Label-free protein quantification using intensity-based Maxquant LFQ values was performed as previously described [29]. Then, high-affinity binders were discriminated against highly abundant low-affinity proteins by competition with the free unmodified compound. Bioinformatic analysis revealed that three proteins showed an increased abundance when treated with various concentrations of SIMR1281 (Appendix A). This study suggested that SIMR1281 binds to proteins involved in the cellular response to oxidative stress, namely, glutathione reductase (GSHR), thioredoxin reductase (TrxR), and inositol-3-phosphate synthase (Figure 2a–c). To confirm whether SIMR1281 maintains this effect in cancer cells, its activity against GSHR and TrxR in HCT116 cells was investigated. As a result, SIMR1281 strongly inhibits GSHR, whereas it moderately inhibits TrxR with IC_50_ values of 0.59 and 2.42 μM, respectively (Figure 2d,e). Since the inhibition of TrxR and GSHR systems is known to affect the cell redox balance, the effect of SIMR1281 on mitochondrial function was investigated. Thus, treating cells with SIMR1281 for 24 h significantly increased redox potential, reduced ATP production, and increased the mitochondrial membrane potential (Figure 2f) [30]. These findings suggest that SIMR1281 has the potential to promote oxidative stress, DNA damage, and apoptosis mechanisms [31,32]. 

### 3.3. SIMR1281 Induces DNA Damage and Regulates Checkpoint Kinases, Ras/ERK and PI3K/Akt Pathways 

The inhibition of TrxR and GSHR systems by SIMR1281 disrupts the cell redox balance and elevates reactive oxygen species levels, leading to the induction of DNA lesions. DNA damage initiates a cascade of signal transduction that involves the activation of ATM and ATR kinases which phosphorylate the H2Ax at serine 139 (γ-H2Ax) and the downstream effector proteins such as p53, checkpoint kinases (Chk2 and Chk1), and apoptosis inducers [33]. To validate this concept, the effect of SIMR1281 on DNA damage response machinery was studied in breast and colon cancer cells by measuring the activation of ATM, ATR, and the downstream checkpoint kinases. The treatment of breast cancer cell lines (MCF7 and SkBr3) and colon cancer cell line (HCT-116) with the IC_50_ and twice the IC_50_ of SIMR1281 for 24 h increased the level of γ-H2Ax and p-ATM, while a minimal effect was detected in the phosphorylation level of ATR (Figure 3a, Appendix A). Strikingly, SIMR1281 treatment induced the phosphorylation of the downstream checkpoint kinases Chk2 and Chk1 in a concentration-dependent manner in both breast cancer cell lines. However, SIMR1281 did not affect the expression level of phosphorylated Chk1 in HCT-116 cells (Figure 3b, Appendix A). In addition, SIMR1281 induced the expression of p53 in a concentration-dependent manner in MCF7 cells, while it induced a significant decrease in p53 expression in SKBr3 (Appendix A). Moreover, SIMR1281 induced the expression of p21 in both breast cancer cell lines (Figure 3b, Appendix A). Furthermore, HCT116 treated with half the IC50 concentration of SIMR1281 showed an increase in p53 expression, while treatment with the IC50 concentration significantly reduced p53 expression. Additionally, treatment with half of the IC50 concentration induced the expression of p21, whereas the treatment of cells with the IC50 concentration decreased p21 expression (Figure 3b). This finding further supports the ability of SIMR1281 to induce DNA damage and to activate DNA damage response machinery. 

Ras/ERK and PI3K/Akt signaling pathways are largely affected by the increase in reactive oxygen species involved in various cellular processes. In addition, they are frequently deregulated in various types of cancers [34,35,36]. Therefore, we examined whether SIMR1281 modulates Ras, ERK, and Akt’s expression in various cancer cell lines. As a result, SIMR1281 reduced the phosphorylation of ERK1/2 in MCF7 and HCT116 cells. However, the treatment of SKBr3 cells with half the IC_50_ concentration did not affect the level of phosphorylated ERK1/2, while treatment with the IC_50_ concentration increased the ERK1/2 phosphorylation (Figure 3c, Appendix A). On the contrary, the phosphorylation of Akt was enhanced in MCF7 and HCT116 cells, while it was reduced in SkBr3 cells following treatment with SIMR1281. These findings are in line with the proteomic data and support the notion that SIMR1281 has a binding affinity towards inositol-3-phosphate synthase in a concentration-dependent manner [37] (Figure 2a,b). These results demonstrate that SIMR1281 regulates Ras/ERK and Akt signaling pathways’ activity, thereby attenuating cancer cell survival, proliferation and enhancing mobility.

### 3.4. SIMR1281 Induces Apoptosis and Cell Cycle Arrest

To investigate whether SIMR1281 induces cell cycle arrest, its effect on MCF7 and HCT116 cells at various time intervals were studied (Figure 3d, Appendix A). These results showed a significant arrest of MCF7 cells at G1/S borders at all time intervals with the appearance of SubG1 fraction of cells at the 72 h time point. However, for HCT116 cells, G1/S arrest was evident at earlier time points (16 and 24 h), whereas an intra- S-phase arrest was seen at later time points (48 and 72 h). Notably, SIMR1281 did not interfere with the cell cycle progression of normal F-180 cells up to 72 h, in contrast to doxorubicin which induced G2/M arrest at all time intervals. To test whether SIMR1281 activates the induction of apoptosis upon the prolonged arrest of cells at different phases of the cell cycle, its effect was evaluated on various cancer cells, including MCF7, SkBr3, and HCT116 (Figure 3e,f, Appendix A). the results revealed a mitochondrial-mediated apoptotic induction, which was indicated by the cleavage of caspase-3 in SKBr3 cells. However, no cleavage was detected in HCT116-treated cells. To further validate the apoptotic mechanism of SIMR1281, its effect on pro-apoptotic proteins, such as Bax and caspase-9, was investigated. Thus, the treatment of MCF7 cells with the IC_50_ concentration of SIMR1281 showed an increase in caspase-9 cleavage. However, no increase in caspase-9 cleavage was detected in HCT116 and SKBr3 cells. Furthermore, Bax expression was increased in HCT116 and MCF7 in a concentration-dependent manner, while in SKBr3, Bax expression was increased only upon treatment with the half IC_50_ concentration (Appendix A). 

It is reported that the overexpression of the oncogenic protein c-Myc results in tumor cell proliferation and migration [38]. Interestingly, SIMR1281 inhibited this oncogene in all tested cell lines (Figure 3f, Appendix A). 

To further confirm the induction of apoptosis by SIMR1281, annexin V staining was studied in MCF7 cells. An increased fraction of late apoptotic cells (annexin V+/propidium iodide+) from 7.5% in control cells to 30.8% in cells treated with the IC_50_ of SIMR1281 was observed (Figure 3g). In contrast, the late apoptotic fraction of normal F-180 cells was not affected by treatment with SIMR1281 using the same concentration. This important finding demonstrates the unique anticancer activity of SIMR1281. Of particular importance are its effects on the oncogenic protein c-Myc, the expression of the tumor suppresser gene, p53, and its downstream target, p21. 

### 3.5. SIMR1281 Disrupts Tubulin Polymerization 

Many anticancer drugs exert their cytotoxic effects by perturbing polymerization dynamics and the depolymerization of microtubules, which is crucial for active mitotic cell division. To test whether SIMR1281 has the ability to disrupt tubulin structure, immunofluorescence staining assay was used (Figure 4a). This study revealed that SIMR1281 inhibited the microtubule kinetics, comparable to Paclitaxel’s (Figure 4b). This finding suggested that SIMR1281 stabilizes microtubule polymerization, thereby arresting cell proliferation in the G2/M phase, as observed in HCT116-treated cells. In agreement with immunofluorescence results, the Western blot analysis of G2/M cyclins revealed that treatment with SIMR1281 reduced cyclin A and cyclin B expression in MCF7 cells (Figure 4c, Appendix A). To confirm this finding, the antiproliferative effect of SIMR1281 was tested on MCF7 cells synchronized at the G2/M phase of the cell cycle since tubulin-targeting drugs exert their effect in this phase (Figure 4d). The percentage of cells at the G2/M phase was 9% in asynchronous cells and was increased to 18% after synchronization (Appendix A). This was accompanied by a reduction in SIMR1281’s IC_50_ from 2.52 µM in asynchronous MCF7 cells to 1.95 µM for G2/M cells (Figure 4e). 

### 3.6. Safety Profile of SIMR1281 in SJL/J Mice 

The observed safety of SIMR1281 in F-180 cells was further confirmed in vivo using SJL/J mice. The administration of a single dose of SIMR1281 (5, 10, 20, 40, 80, and 160 mg/kg) was well-tolerated without overt signs of toxicity or weight loss (Figure 5a). Furthermore, in a multiple-dose study, mice received daily doses of vehicle or SIMR1281 (25, 50 mg/kg) for 14 days. Similarly, no signs of toxicity or weight loss were noticed (Figure 5a). Moreover, no histologic lesions consistent with toxic injury were seen in any microscopically examined organs, including the heart, kidney, and liver (Figure 5b,c). Hematology and serum chemistry analysis following the treatment of subcutaneous HCT116 xenograft tumors with SIMR1281 for 30 days revealed normal values of all tested parameters compared to the control group. A mild reduction in red blood cell number and albumin levels was noticed (Appendix A). These findings support the notion that SIMR1281 has a high in vivo safety profile.

### 3.7. SIMR1281 Significantly Attenuates the Proliferation of HCT116 Xenograft Model

To determine whether the SIMR1281 anticancer phenotype manifests in vivo, a colon xenograft in mice was established. The HCT116 xenograft mice model was treated with 50 mg/kg of SIMR1281 for 30 days. Mice weight and tumor volume were monitored throughout the study. No significant change in mice weight was observed in treated groups compared to the vehicle-treated group (Figure 5d). Tumor volume was measured twice a week using a digital caliper, which revealed a significant reduction in HCT116 xenograft tumor proliferation in SIMR1281-treated groups relative to vehicle-treated groups (Figure 5e,h). At the end of the study, mice were sacrificed, and tumors were collected. A significant reduction in tumor weight was observed (Figure 5g). Additionally, immunohistochemistry analysis was conducted for the detection of the expression of the proliferation marker Ki67. The mean of Ki67-expressing cells from three representative fields per section per sample indicated a significant decrease in Ki67 expression in the 50 mg/kg dose of SIMR1281-treated group (Figure 5f). These findings further confirm the potent effect of SIMR1281 as a novel anticancer modality and provide a concrete corroboration that SIMR1281 effectively disrupts the cancer ecosystem both in vitro and in vivo.

## 4. Discussion 

The rising incidence rate of cancer resistance and harsh side effects of existing anticancer medications highlights the pressing need to develop new multitarget therapies that selectively disrupt cancer cells while sparing their normal counterparts. Many of the marketed anticancer drugs are single target-based drugs. This model oftentimes is becoming less effective in the management of complex diseases such as cancer [1,2]. As an alternative approach, drug combination therapy was employed. However, drug combination is associated with an array of adverse side effects [3,4]. To address these obstinate obstacles, efforts were directed towards discovering drugs with a "multitarget" mechanism of action [5,6]. Borrowing inspiration from these efforts, our developed compound library was subjected to phenotypic screening against a panel of cancer cell lines [39]. We report herein a novel chemical probe SIMR1281 that possesses a potent anticancer activity. The compound showed potent antiproliferative activity against a wide range of cancer cells from different histological and genetic backgrounds and belonged to both liquid and solid malignancies. In addition, SIMR1281 was also found to be equally effective against resistant MCF7 and A549 cells.

To uncover the mechanism of action of SIMR1281, proteomic analysis was performed. This study revealed that SIMR1281 modulates the functions of thioredoxin reductase 1, glutathione reductase, and inositol-3-phosphate synthase, key enzymes that are overexpressed in cancer cells. Higher levels of Trx and GSH were also reported to contribute to resistance to anticancer therapy and aggravate the cancer heterogeneity problem. Currently, there are no clinical anticancer drugs that specifically target Trx and GSH redox systems. However, many clinical drugs were reported to inhibit TrxR with varying potency, albeit with high toxicity [40,41,42]. One hurdle in developing successful TrxR inhibitors is their cross-reactions with other cellular thiols [40]. Unlike those metal complexes and Michael acceptors known to inhibit these enzymes, SIMR1281 lacks reactive groups (such as selenol or sulfhydryl group) known to undergo cross-reactivity with cellular thiol groups, which led to their failure in clinical trials. To further validate the enzymatic activity of SIMR1281 against TrxR and GSHR, cell-based assays revealed that SIMR1281 inhibits TrxR and GSHR with IC_50_ values of 2.42 µM and 0.59 µM, respectively [43].

Several cancer cells produce large amounts of ROS and elevate the antioxidant defense mechanisms, mainly through the upregulation of the Trx and GSH systems [30]. The inhibition of TrxR by SIMR1281 led to an elevated level of reactive oxygen species (ROS), which aided in cell apoptosis and prevented cell proliferation. The exposure of cancer cells to SIMR1281 inactivates both Trx and GSH systems, severely compromising the antioxidant capacity of cells, leading to a marked increase in ROS, which aided in cell apoptosis and prevented cell proliferation. Furthermore, SIMR1281 significantly increased cell redox potential, reduced ATP content, and increased mitochondrial membrane potential. These effects are known to suppress cell proliferation and promote cell death.

Deregulated inositol metabolism has been observed in several diseases that modulate various pathways responsible for cancer progression [44,45]. SIMR1281 has shown a concentration-dependent affinity toward the inositol-3-phosphate synthase enzyme in the DARTS proteomic assay. This finding was supported by the reduced activity of Akt in SkBr3 cells when treated with SIMR1281 [46]. 

An increased intracellular ROS level is associated with the induction of different DNA damage lesions [47]. The phosphorylation of H2Ax and activation of ATM, Chk1, Chk2, p53, and p21 after treatment with SIMR1281 confirms the induction of DNA damage following ROS formation due to the dual inhibition of TrxR and GSHR by SIMR1281. This finding is consistent with the observed arrest of cells at G1/S, intra-S, and G2/M phases. Blocking the cell cycle progression of cells with damaged DNA allows cells to repair their DNA preceding entry into the critical S- and M-phases [48]. When cells fail to repair the damaged DNA, they are permanently arrested at one phase of the cell cycle or stimulate different death pathways, including apoptosis [49]. SIMR1281 triggers the induction of pro-apoptotic proteins such as caspase-3, caspase-9, and BAX while it down-regulates growth-stimulating proteins such as c-Myc. These effects indicate cell apoptosis and explain the cell death observed when cells were treated with SIMR1281. These events were not observed in the normal F-180 cells following treatment with SIMR1281, demonstrating its safety profile. 

Ras/ERK and PI3K/Akt signaling pathways are hyperactivated in many human malignancies, and their activity is modulated by the level of intracellular ROS [50]. One of the novel mechanisms is how SIMR1281 exerts its multitarget anticancer activity in regulating Ras/ERK and PI3K/Akt signaling pathways. SIMR1281 induced a reduction in the phosphorylation of ERK1/2 in MCF7 and HCT116 cells, whereas it reduced the activity of Akt in SkBr3 cells. However, the activity of ERK1/2 in SkBr3 and the phosphorylation of Akt in MCF7 and HCT116 cells were increased when treated with SIMR1281. This divergent effect of SIMR1281 on ERK1/2 and Akt might be due to the cross-inhibition between Ras/ERK and Akt pathways, which are known to deleteriously impact the activity of each other [51]. These findings were supported by the proteomic study, in which SIMR1281 was found to possess a high affinity to inositol-3-phosphate synthase, an upstream event of Akt signaling [37,52]. 

These promising findings encouraged us to examine the in vivo safety profile of SIMR1281 in mice. SIMR1281 was indicated to be safe in mice treated with up to 150 mg/kg, without any overt signs of toxicity or weight loss (Figure 5a). To further validate the preclinical anticancer effect of SIMR1281, an in vivo efficacy study in nude mice bearing human HCT116 cells xenografts was conducted. Mice treated once a day for 30 days with SIMR1281 (50 mg/kg, i.p.) indicated a significant decrease in tumor volume compared to the vehicle (Figure 5e–h) without any signs of toxicity or changes in mice weight relative to the vehicle (Figure 5b, Appendix A). Additionally, staining HCT116 xenograft for the proliferation marker ki67 [53] revealed a significant reduction in tumor proliferation after treatment with SIMR1281 (Figure 5f). The promising in vivo anticancer activity of SIMR1281 forms the foundation for further development into preclinical and clinical studies. 

Collectively, SIMR1281 has shown promising anticancer activity in vitro against multiple types of cancers and in vivo against HCT116 xenograft model while maintaining a high in vivo safety profile. Additionally, SIMR1281 targets multiple enzymes overexpressed in cancer cells. This unique activity makes SIMR1281 a first-in-class molecule for cancer chemotherapy. Furthermore, the novel multitarget mechanism of action of SIMR1281 and its observed safety profile may curtail the drawbacks encountered by marketed anticancer drugs. 

## 5. Conclusions

The development of multitarget anticancer drugs is currently the optimal approach followed by researchers to attain efficacy and overcome the challenges encountered by the traditional treatment approaches. To this end, we have identified a novel anticancer small molecule, SIMR1281, that possesses a multitarget mechanism of action. SIMR1281 has shown promising anticancer activity in vitro and in vivo against multiple types of cancers while maintaining a high in vivo safety profile. In addition, SIMR1281 targets multiple enzymes overexpressed in cancer cells. This unique activity makes SIMR1281 a first-in-class molecule for cancer chemotherapy. Furthermore, the disruptive pleiotropic mechanism of SIMR1281 on the cancer microenvironment represents an innovative strategy for the treatment of various types of cancers and places this compound in a privileged position as a potential lead drug candidate.

## 6. Patents

Patent application (T. H. Al-Tel, Raafat A. El-Awady, Srinivasulu Vunnam, Cijo G. Vazhapily, Hany A. Omar, Novel heterocyclic systems and pharmaceutical compositions thereof. US Patent No US20190292204A1).

## Figures and Tables

**Figure 1 cancers-13-02840-f001:**
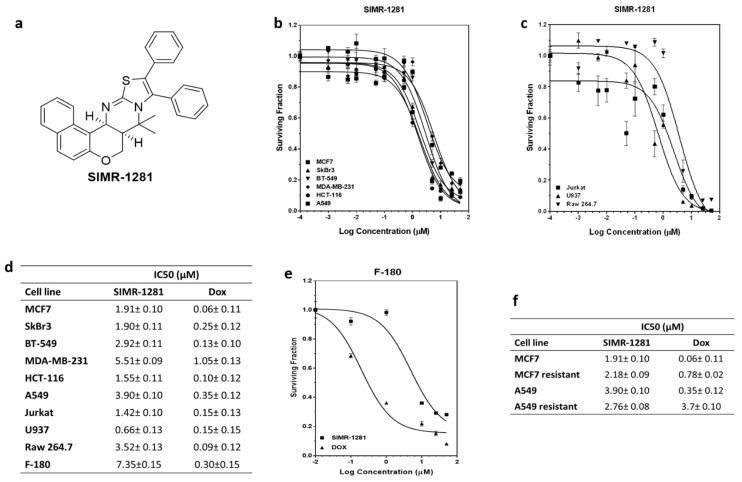
The antiproliferative effect of SIMR1281 on a panel of cancer cell lines and normal fibroblast cells (F-180). (**a**) Chemical structure of SIMR1281. (**b**,**c**) Dose-dependent effect of SIMR1281 on the indicated cell lines. (**d**) The IC_50_s of SIMR1281 in cancer cell lines and in normal F-180 cells, Data are mean ± SEM (*n* = 3). (**e**) Surviving fraction of SIMR1281 and doxorubicin-treated F-180 cells. (**f**) The IC_50_ of SIMR1281 in MCF7 and A549 cancer cell lines that acquired resistance to cancer medications, data are mean ± SEM (*n* = 3).

**Figure 2 cancers-13-02840-f002:**
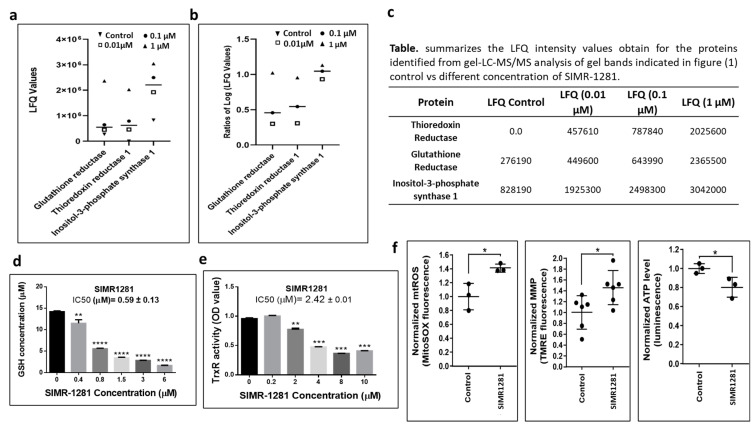
SIMR1281 targets the function of key proteins and modulates mitochondrial functions. (**a**) the LFQ ratios of cell lysate treated with different concentrations as indicated. (**b**) The ratios (treatment/control) of transformed LFQ values. (**c**) A table summarizes the LFQ intensity values for the identified proteins. (**d**,**e**) The effect of SIMR1281 on the concentration of oxidized glutathione (GSH) and thioredoxin reductase activity. Columns are mean ± SD (*n* = 3). (**f**) Cellular redox potential, MMP, and ATP level upon the treatment with SIMR1281. *p*-value determined by two-tailed unpaired *t*-test. * indicates significant difference versus control at *p* < 0.05 (*), *p* < 0.01 (**), *p* < 0.001 (***), *p* < 0.0001 (****).

**Figure 3 cancers-13-02840-f003:**
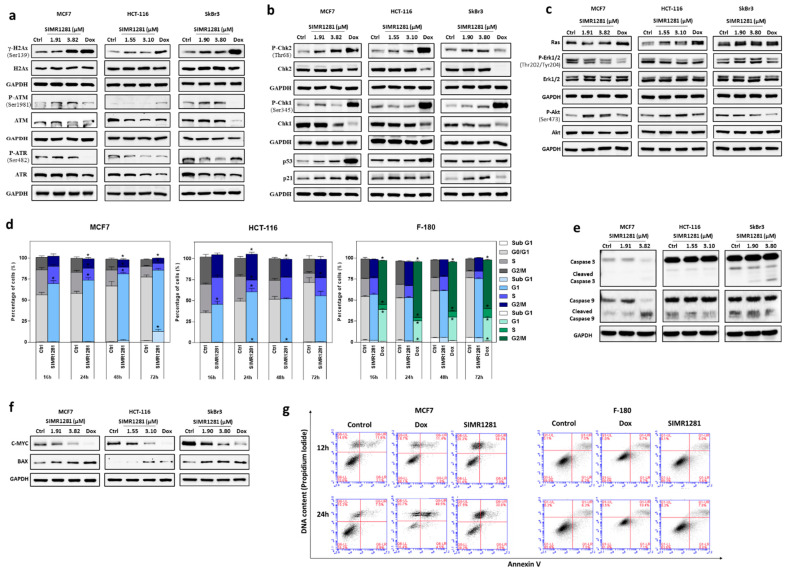
SIMR1281 possesses a potent pleiotropic anticancer effect. (**a**–**c**) Western blot analysis of the expression levels of the indicated proteins following treatment with SIMR1281 in MCF7, HCT-116, and SkBr3 cells (*n* = 3). (**d**) Cell cycle arresting potentials of SIMR1281 in the treated MCF7, HCT-116 and F-180 cells (*n* = 3). * Indicates significant difference versus control at *p* < 0.05 (*) by two-tailed unpaired *t*-test. (**e**,**f**) Immunoblots of the indicated proteins following treatment with SIMR1281 in MCF7, HCT-116, and SkBr3 cells (*n* = 3). (**g**) Annexin V/PI staining of MCF7 and F-180 cells after treatment with Dox and SIMR1281 (*n* = 3). Full Western blots are available in Appendix A,S7,S8.

**Figure 4 cancers-13-02840-f004:**
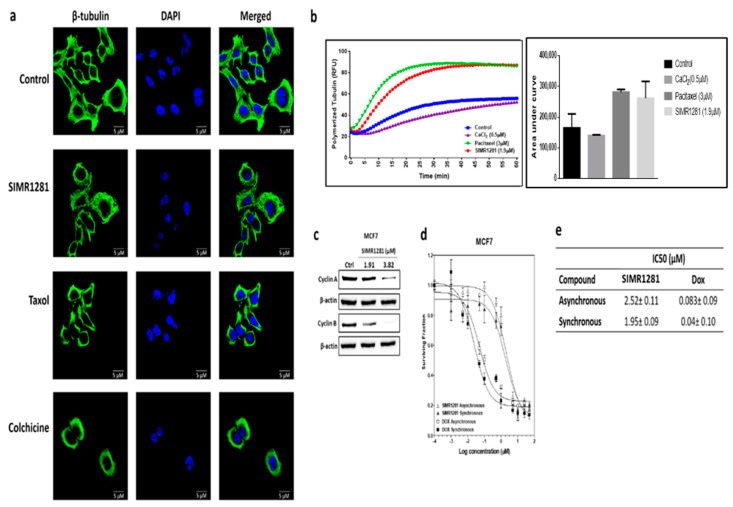
SIMR1281 stabilizes microtubule polymerization. (**a**) Immunofluorescence analysis of tubulin polymerization in A549 cell line after 24 h of treatment with the IC_50_ concentration of SIMR1281, taxol, and colchicine (scale bar 5 µm). (**b**) Tubulin polymerization kinetic effect of the treatment with the IC_50_ concentration of SIMR1281. (**c**) Western blot analysis of cyclins A and B expression levels following treatment with SIMR1281 in MCF7 cells. (**d**) Dose-dependent effect and IC_50_ values of SIMR1281 and doxorubicin treatment in asynchronous and synchronous MCF7 cells. (**e**) The IC_50_ of SIMR1281 in asynchronous and synchronous MCF7 cells. Data; mean ± SEM (*n* = 3). Full Western blots are available in Appendix A.

**Figure 5 cancers-13-02840-f005:**
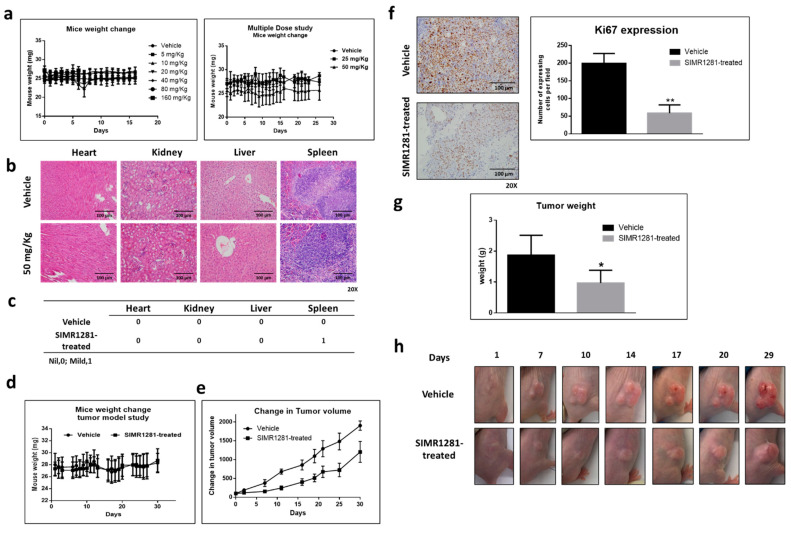
SIMR1281 is safe at high doses, and it significantly attenuated the proliferation of HCT116 tumor model in nude mice. (**a**) Mice weight change over the indicated period during the treatment in single- and multiple-dose studies of SIMR1281. Mean; bars, SD (*n* = 3). (**b**) Representative figures of H&E staining (scale bar 100 µm) for SIMR1281-treated (50 mg/kg) and control mice groups. (**c**) Table displaying the toxicity score for SIMR1281-treated (50 mg/kg) and control mice groups. (**d**) Mice weight change during treatment. Mean; bars, SD (*n* = 5–7). (**e**) Change in tumor volume throughout the 30 days of treatment with SIMR1281. Mean; bars, SD (*n* = 5–7). (**f**) The mean of Ki67 expressing cells. Representative pictures of IHC for Ki67 expression in the tumor tissues (scale bar 100 µm) after the treatment with 50 mg/kg of SIMR1281. Columns, mean; bars, SD (*n* = 5–7). ** Indicates a significant difference versus control at *p* < 0.05 determined by a two-tailed unpaired *t*-test. (**g**) Tumor weight of the vehicle and SIMR1281-treated group. * indicates a significant difference versus control at *p* < 0.01. (**h**) Representative pictures of tumor progression in SIMR1281-treated and control groups.

## Data Availability

Not applicable.

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
