# Peer review of "A Novel Benzopyrane Derivative Targeting Cancer Cell Metabolic and Survival Pathways"

_cancers, 2021, doi:10.3390/cancers13112840_

Round 1
Reviewer 1 Report
This manuscript is very elegantly written, meticulously designed, very well described and discussed.
It carries all the merit for the acceptance, but needs some minor corrections before acceptance.
Some minor comments are....
1) Line 62 - change "multi-product drugs" to "multi-target drugs"
2) Line 87 - the word 'spread' is repeated, delete one.
Or
replace " biomolecules needed for cancer survival, proliferation, replication, spread and spread" with "biomolecules needed for cancer survival, growth and spread".
3) In many places typological/ graphical errors need to be corrected.
Line 109 - 37 oC - the degree sign should be in superscript.
Line 110- CO2 - '2' should be in superscript.
Line 117, 161, 164, 171, 182, 307 and may be some other places, please change the last number in "1x106 or 2x106 or 3x106" to the power of 6 (superscirpt).
4) In lines 341-343, the sentence can be rewritten as " Among the structural options, the benzopyrane scaffold is a privileged architecture representing the core structure of a wide range of biologically appealing molecules".
Author Response
This manuscript is very elegantly written, meticulously designed, very well described, and discussed.
It carries all the merit for the acceptance but needs some minor corrections before acceptance.
Some minor comments are....
We really appreciate your supportive and constructive comments, which significantly improved the manuscript.
- Line 62 - change "multi-product drugs" to "multi-target drugs"
Response:
As per the reviewer's comment, this was corrected.
- Line 87 - the word 'spread' is repeated, delete one.Or replace " biomolecules needed for cancer survival, proliferation, replication, spread and spread" with "biomolecules needed for cancer survival, growth and spread".
Response:
All corrections were made as directed.
- In many places typological/ graphical errors need to be corrected.
Line 109 - 37 oC - the degree sign should be in superscript.
Line 110- CO2 - '2' should be in superscript.
Line 117, 161, 164, 171, 182, 307 and may be some other places, please change the last number in "1x106 or 2x106 or 3x106" to the power of 6 (superscirpt).
Response:
We appreciate the reviewer's keen observation. All corrections were done and highlighted in the manuscript.
- In lines 341-343, the sentence can be rewritten as " Among the structural options, the benzopyrane scaffold is a privileged architecture representing the core structure of a wide range of biologically appealing molecules".
Response:
The sentence was rephrased as suggested.
Reviewer 2 Report
To Authors:
Dear Authors,
The article “A Novel benzopyrane derivative targeting cancer cell metabolic and survival pathways'' by D.M. Zaher et al., aimed to characterize synthetic substance SIMR1281 (novel benzopyrane derivative) as a potential novel tool for cancer treatment. The authors propose that synthetic compound based on benzopyrane possesses activity towards glutathione reductase (GSHR), thioredoxin reductases (TrxR), mitochondrial metabolism, DNA damage, cell cycle progression, as well as activity a substantial anti-proliferative effect and induction of apoptosis on tested cancer cell lines. The work performed in this paper used cell lines representing the common types of cancer: Human acute T cell leukemia cell line (Jurkat), Abelson murine leukemia cell line (Raw 246.7), Human histiocytic lymphoma cell line (U937), Human colorectal carcinoma cell line (HCT116), Human Non-small cell lung cancer cell line (A549) and human breast carcinoma cell lines (MCF-7, MDA-MB231), as well as normal human dermal fibroblasts cell line (F180). Experiments and assays used in the current study are up-to-date and well established. The study is not limited by in vitro experiments and is supplemented by biochemical assays and animal studies that add significant value to this work.
In general paper creates a positive impression despite this, it requires moderate stylistic and editorial changes. The central idea of current study is very interesting and well characterized. According to data presented on figures assays performed correctly using the appropriate controls.
I have suggested several key questions that the authors should answer to be accepted.
Main concerns and comments:
Question 1. Despite Figure 1D demonstrates that IC50 values of studied compounds area much higher in F180 line compare to cancer lines, I would recommend adding cell cycle profiles and 7AAD/Annexin plots with F180 after control/dox/ SIMR1281 treatment (in same concentration as for Figure 3F, at least in supplementary data to convince readers that this effect is mostly realized on cancer lines.
Question 2. Is there any data confirming that the compound does interfere with tubulin polymerization/depolymerization to fully realize their activity toward cell cycle progression (G2/M) except polymerization assay? It is not clear that this effect causes G2/M arrest because in Figure 3C there is no increase of cell number in columns (G2/M) at 24,48,72 hours of treatment. As a solution I would recommend combining cell cycle profiling with antibody staining for mitotic markers (H3 phosphorylation or cyclins) and perform the experiment with positive control (paclitaxel). Please correct me if I misunderstand the presented data, but according to the data shown on Figure 3C it is obvious that there is no substantial G2/M arrest.
Question 3. I would suggest repeating the cell cycle profiling experiment due to its limited informative value (Figure 3C). Even without treatment, the cell cycle profile changes dramatically in control samples (for example at 24h G1: about 50% while at 72h G1:75%). Is it the effect of overgrowing culture?
Question 4. If compound promote oxidative stress and subsequent cell cycle arrest in G1 it would be nice to add direct markers for DNA damage (as yH2AX and pATM/ATR)
Question 5. Keeping in mind that effect of SIMR1281 is mainly realized through cellular targets as Glutathione reductase (GSHR), Thioredoxin reductase (TrxR) and inositol-3-phosphate synthase, I would suggest evaluating expression levels of that proteins in tested cell lines and probably there is some link between expression levels and sensibility to drug treatment.
Minor concerns and comments.
Line: 381 using a (28) (missing word)?
Figure 3F. It is a minor suggestion to put DNA content on Y-axis and Annexin V scale on X-axis (this way of data presentation is more common when apoptotic event on the two right quadrants (lower represents early and upper - late apoptosis). I would suggest adjusting compensation in PI and Annexin V channels, they should have close values on PI channels (because membrane at early apoptosis should not be permeable for propidium iodide). Number of events in dox 24 hours is very low, I believe that increasing the number of events could serve as a milestone to distinguish early/late apoptotic gates.
Figure 4. Scale bar is missing for IF images. Legends should contain precise descriptions of data shown. For example: “Fig. 4 SIMR1281 stabilizes microtubule polymerization. a, Immunofluorescence analysis of tubulin polymerization in … cell line after … hours of treatment with ….. (scale bar 1um); b, Tubulin polymerization kinetic effect of the treatment with …….; c, Dose-dependent effect and IC50 values of SIMR1281 and doxorubicin treatment in asynchronous and synchronous MCF7 cells. “.
Figure5. It would be nice to put figures in logic order. (from up-to-down). Figures 5B, F scales bars are missing. It would be nice to rename treated samples to “SIMR1281-treated”and remove the title “50 mg/kg” from images.
Line 307: probably “FBS free media”
Line 482: SEM is mentioned in legend, not present on figure
Line 516: probably red blood cell number
Author Response
In general paper creates a positive impression despite this, it requires moderate stylistic and editorial changes. The central idea of current study is very interesting and well characterized. According to data presented on figures assays performed correctly using the appropriate controls.
I have suggested several key questions that the authors should answer to be accepted.
Main concerns and comments:
We really appreciate your supportive and constructive comments..
- Despite Figure 1D demonstrates that IC50 values of studied compounds area much higher in F180 line compare to cancer lines, I would recommend adding cell cycle profiles and 7AAD/Annexin plots with F180 after control/dox/ SIMR1281 treatment (in same concentration as for Figure 3F, at least in supplementary data to convince readers that this effect is mostly realized on cancer lines.
Response:
Thank you very much for the great suggestions that would enhance the quality of the manuscript. The cell cycle and Annexin V/PI experiments were performed for F-180 cells and were added to figure 3 (Fig. 3d & g) and were described in the "results" section.
- Is there any data confirming that the compound does interfere with tubulin polymerization/depolymerization to fully realize their activity toward cell cycle progression (G2/M) except polymerization assay? It is not clear that this effect causes G2/M arrest because in Figure 3C there is no increase of cell number in columns (G2/M) at 24,48,72 hours of treatment. As a solution I would recommend combining cell cycle profiling with antibody staining for mitotic markers (H3 phosphorylation or cyclins) and perform the experiment with positive control (paclitaxel). Please correct me if I misunderstand the presented data, but according to the data shown on Figure 3C it is obvious that there is no substantial G2/M arrest.
Response:
Thank you for addressing this point. The western blot analysis was performed in MCF7 cells treated with SIMR1281 for detecting the expression of cyclin A and B, which are involved in G2/M transition. These results were added to figure 4 (Fig. 4c) and were described in the "results" section.
- I would suggest repeating the cell cycle profiling experiment due to its limited informative value (Figure 3C). Even without treatment, the cell cycle profile changes dramatically in control samples (for example, at 24h G1: about 50% while at 72h G1:75%). Is it the effect of overgrowing culture?
Response:
Thank you for your comment. Yes, this time-dependent change of fraction of control cells in G1 phase is due to overgrowing cells, and to cancel this effect, we used a specific control for each time point.
- If the compound promotes oxidative stress and subsequent cell cycle arrest in G1 it would be nice to add direct markers for DNA damage (as yH2AX and pATM/ATR)
Response:
Thank you for this suggestion. The western blot data for p-H2Ax, p-ATM, and p-ATR in MCF7, SkBr3, and HCT-116 cells treated with SIMR1281 has been added to figure 3 (Fig. 3a). In addition, the full blots and band quantification have been provided in the supplementary file.
- Keeping in mind that effect of SIMR1281 is mainly realized through cellular targets as Glutathione reductase (GSHR), Thioredoxin reductase (TrxR) and inositol-3-phosphate synthase, I would suggest evaluating expression levels of that proteins in tested cell lines and probably there is some link between expression levels and sensibility to drug treatment.
Response:
Thank you for raising this important point. We totally agree with the reviewer's comment. As determined by the DARTS assay, SIMR1281 has a high affinity to Glutathione reductase (GSHR), Thioredoxin reductase (TrxR) and inositol-3-phosphate synthase, which suggests its direct effect on their activities and functions. To assess this, enzymatic assays were performed and confirmed SIMR1281's effect in modulating TrxR and GSH. Furthermore, the expected downstream effects, as the production of ROS was explored and confirmed. Testing SIMR1281 effect on the expression levels of these proteins would give further insight into the anticancer activity of SIMR1281. However, the determined enzymes' activity modulation would be the primary direct effect, while the change in the protein expression would be a secondary result.
- Line: 381 using a (28) (missing word)?
Response:
Per the reviewer's comment, this was corrected.
- Figure 3F. It is a minor suggestion to put DNA content on Y-axis and Annexin V scale on X-axis (this way of data presentation is more common when apoptotic event on the two right quadrants (lower represents early and upper - late apoptosis). I would suggest adjusting compensation in PI and Annexin V channels, they should have close values on PI channels (because membrane at early apoptosis should not be permeable for propidium iodide). Number of events in dox 24 hours is very low, I believe that increasing the number of events could serve as a milestone to distinguish early/late apoptotic gates.
Response:
Thank you for this observation. After checking the raw data of Annexin V/PI staining, we have realized that the labels for the x-axis and y-axis were presented oppositely. Therefore, we have corrected the labels according to the raw data file (Figure 3g). Additionally, we have made minor modifications in compensation for PI and Annexin V channels. The compensation was adjusted using the single-stained samples ( Annexin V only and PI only) following the guidelines of Accuri C6 software. In the case of Dox 24hours samples, we have increased the number of events as much as possible.
- Figure 4. Scale bar is missing for IF images.
Legends should contain precise descriptions of data shown. For example: "Fig. 4 SIMR1281 stabilizes microtubule polymerization. a, Immunofluorescence analysis of tubulin polymerization in … cell line after … hours of treatment with ….. (scale bar 1um); b, Tubulin polymerization kinetic effect of the treatment with …….; c, Dose-dependent effect and IC50 values of SIMR1281 and doxorubicin treatment in asynchronous and synchronous MCF7 cells. ".
Response: We thank the reviewer for his suggestion. The scale bar for immunofluorescence was added, the legends were revised, and precise descriptions were added.
- It would be nice to put figures in logic order. (from up-to-down)
Figures 5B, F scales bars are missing.
It would be nice to rename treated samples to "SIMR1281-treated" and remove the title "50 mg/kg" from images.
Response: We appreciate the reviewer's comment. The figures in Figure5 were reordered as suggested, and 50 mg/kg was replaced with SIMR1281-treated. In addition, the scale bars for Figures 5B, F were added as per the reviewer's request.
- Line 307: probably "FBS free media"
Response: Thank you for your observation. It was corrected.
- Line 482: SEM is mentioned in the legend, not present on the figure
Response: Per the reviewer's comment, this was corrected.
- Line 516: probably red blood cell number
Response: Thank you for your observation; it was corrected.
Reviewer 3 Report
The authors report the discovery of mechanism of action involving multi-targets and in vitro/in vivo anticancer effect of a novel benzopyrane derivative. The manuscript is well written. The research results are very interesting to readers and supported with strong experimental data. Overall, the manuscript is acceptable for publication with minor revisions.
Some specific comments and suggestions are as follows:
1. SIMR1281 was used in the text, but SIMR-1281 was seen in Figures and figure legends. Keep the compound name consistent throughout the manuscript.
2. Page 2, line 46, the authors claim the compound significantly reduced tumor volume. But the volume difference between the treatment and control groups in Fig 5E is not that big, less than 50% reduction at day 30 (end of the study) even at 50 mg/kg dose. The potency of the compound alone is moderate. This reviewer suggests removing the word “significantly”.
3. Page 1, line 32, and Page 2, line 49, current data are not sufficient to support the compound to be evaluated in a clinical setting. This review suggests changing the sentence to “These results support further pre-clinical development of SIMR1281 as it represents a potential lead compound for cancer treatment.”
4. Page 2, line 62, change “multi-product drugs” to “multi-target drugs”.
5. Page 3, line 117, cell number should be “1×104” in “cells were seeded at a density of 1×104 per well”. The authors need make similar “superscript” correction throughout the manuscript, e.g., p4 line 161, p5 line 164, 171, 182, p8 line 307
6. Page 6, line 231, delete the period “.” before “Streptomycin”.
7. The compound SIMR1281 is quite hydrophobic. On page 8, the authors used formulation of “0.5% DMSO in PBS” for in vivo safety studies even at the dose of 80 and 160 mg/kg, however, they changed to use formulation of “40% PEG400+4% DMSO+PBS” for efficacy study at 50 mg/kg. The reviewer is curious about the solubility of the compound in 0.5% DMSO/PBS at the dose of 80 and 160mg/kg. Please clarify.
8. Page 10, line 366, add abbreviation Dox after doxorubicin since Dox was used in Fig 1d, e f.
9. Page 10, line 381, the sentence is not truncated. Remove “using a”?
10. Page 11, increase font size in Fig 2a, b, c. They are barely readable.
11. Page 15, line 508, is it a single dose? If so, why daily? Please clarify.
12. Page 17, line 576, give full name of “reactive oxygen species” before “ROS”.
13. Page 18, line 591-593, revise the sentence, e.g., add “by” before “SIMR1281”.
14. Page 18, line 618, add the duration of treatment in the sentence.
15. Page 19, line 625-626 and 636, please rewrite the sentence. In vitro data demonstrated the compound is active against a panel of cancer cell lines, however, this manuscript only reports in vivo efficacy study in one tumor model, it is not appropriate to state “in vitro and in vivo against multiple types of cancers”.
16. Page 19, line 642, again, it does not sound good to say the compound is a clinical lead drug candidate. Suggest delete the word “clinical”.
Author Response
Reviewer 3
- SIMR1281 was used in the text, but SIMR-1281 was seen in Figures and figure legends. Keep the compound name consistent throughout the manuscript.
Response:
We appreciate the reviewer's keen observation. All corrections were done and highlighted in the manuscript.
- Page 2, line 46, the authors claim the compound significantly reduced tumor volume. But the volume difference between the treatment and control groups in Fig 5E is not that big, less than 50% reduction at day 30 (end of the study) even at 50 mg/kg dose. The potency of the compound alone is moderate. This reviewer suggests removing the word "significantly".
Response:
We thank the reviewer for raising this comment, and we totally agree with his suggestion, so the word "significantly" was removed.
- Page 1, line 32, and Page 2, line 49, current data are not sufficient to support the compound to be evaluated in a clinical setting. This review suggests changing the sentence to "These results support further pre-clinical development of SIMR1281 as it represents a potential lead compound for cancer treatment."
Response:
Per the reviewer's comment, the sentence was rephrased.
- Page 2, line 62, change "multi-product drugs" to "multi-target drugs".
Response:
Per the reviewer's comment, this was corrected.
- Page 3, line 117, cell number should be "1×104" in "cells were seeded at a density of 1×104 per well". The authors need to make similar "superscript" correction throughout the manuscript, e.g., p4 line 161, p5 line 164, 171, 182, p8 line 307
Response:
We appreciate the reviewer's keen observation. All corrections were done and highlighted in the manuscript.
- Page 6, line 231, delete the period "." before "Streptomycin".
Response: The period was deleted. Thank you.
- The compound SIMR1281 is quite hydrophobic. On page 8, the authors used the formulation of "0.5% DMSO in PBS" for in vivo safety studies, even at the dose of 80 and 160 mg/kg. However, they changed to use the formulation of "40% PEG400+4% DMSO+PBS" for efficacy study at 50 mg/kg. The reviewer is curious about the solubility of the compound in 0.5% DMSO/PBS at the dose of 80 and 160 mg/kg. Please clarify.
Response: In the single dose safety study, the dose of 80 or 160 mg/kg was admisinistered once only and the injection was applied directly after solubilizing SIMR1281 in DMSO and warm PBS, to avoid any precipitation during the preparation or even the precipitation in vivo upon injection. However, for the efficacy study, the dose was administered daily for 30 days, which demands a dissolving formula that assures that SIMR1281 will not precipitate in vivo, thus "40% PEG400+4% DMSO+PBS" formula was used.
- Page 10, line 366, add abbreviation Dox after doxorubicin since Dox was used in Fig 1d, e f.
Response: Thank you, the abbreviation was added.
- Page 10, line 381, the sentence is not truncated. Remove "using a"?
Response: "using a" was removed.
- Page 11, increase font size in Fig 2a, b, c. They are barely readable.
Response: We thank the reviewer for raising this point. The font size was increased.
- Page 15, line 508, is it a single dose? If so, why daily? Please clarify.
Response: We appreciate the reviewer's keen observation. It is a single dose, as described in the methods section. Thus the "daily" word was deleted. Thank you.
- Page 17, line 576, give full name of "reactive oxygen species" before "ROS".
Response: The full name was added.
- Page 18, line 591-593, revise the sentence, e.g., add "by" before "SIMR1281".
Response: The sentence was revised and corrected as suggested.
- Page 18, line 618, add the duration of treatment in the sentence.
Response: Thank you for your comment. The treatment duration was added.
- Page 19, lines 625-626 and 636, please rewrite the sentence. In vitro data demonstrated the compound is active against a panel of cancer cell lines. However, this manuscript only reports in vivo efficacy study in one tumor model, it is not appropriate to state "in vitro and in vivo against multiple types of cancers".
Response: The sentence was revised as suggested and corrected accordingly.
- Page 19, line 642, again, it does not sound good to say the compound is a clinical lead drug candidate. Suggest deleting the word "clinical".
Response: We agree with the reviewer's comment, and the "clinical
Round 2
Reviewer 2 Report
To Authors:
Dear Authors,
I greatly appreciate the time and effort that the authors invested to answer comments and questions from the first round of revision.
Most of the concerns and questions were addressed and satisfactory information have been provided.
Nevertheless there are some points that need to be clarifyed .
Figure 3d. I would suggest changing the color legend for cell cycle columns to make it easier to distinguish cell cycle profiles for each tested condition. As part of curiosity I would like to ask one technical question regarding cell cycle analysis protocol. Line 184. Did Authors collect floating cells separately before and washing with PBS and treatment with trypsin and later combined for staining? Because in most of the cases cells that are in G2/M and M-phases are non-adherent and could be lost during the washing process if the supernatant is not collected separately and spinned. If the experiment is done in a technically correct way, then there is no obvious G2/M arrest (paxlitacel could be used as positive control as a drug causing Mitotic catastrophe).
Figure 3g. I'm wondering why early apoptotic events (PI-/Annexin V+) are underrepresented in PI/Annexin staining after Doxorubicin treatment. Relatively low events in the right lower quadrant in treated samples and unproportionally high events for (PI+/Annexin V-) and late apoptotic (PI+/Annexin V+ ) events in control samples, that value in untreated lines usually should not exceed 2-3%.
Figure 4b. The data is very interesting. I would suggest changing the color (to make them in different colors) of curves in the tubulin polymerization plot to clarify for readers the effect of the studied compound.
Author Response
Reviewer #2
I greatly appreciate the time and effort that the authors invested to answer comments and questions from the first round of revision. Most of the concerns and questions were addressed and satisfactory information has been provided.
Response: Thank you so much for your positive comment. We really appreciate your professional comments that greatly improved our work.
Comment 1. Figure 3d. I would suggest changing the color legend for cell cycle columns to make it easier to distinguish cell cycle profiles for each tested condition. As part of curiosity I would like to ask one technical question regarding cell cycle analysis protocol. Line 184. Did Authors collect floating cells separately before and washing with PBS and treatment with trypsin and later combined for staining? Because in most of the cases cells that are in G2/M and M-phases are non-adherent and could be lost during the washing process if the supernatant is not collected separately and spinned. If the experiment is done in a technically correct way, then there is no obvious G2/M arrest (paxlitacel could be used as positive control as a drug causing Mitotic catastrophe).
Response: Thank you for your suggestion. The colour of cell cycle columns has been changed as suggested.
The cell cycle analysis has been done only for adherent cells. This is because we did not observe a higher fraction of floating cells in treated samples over the control. The low G2/M fraction after treatment with SIMR1281, despite of the effect that was seen in tubulin polymerization experiment and cyclins expression, may be explained on the basis that inhibition of tubulin polymerization is one of the multiple effects of SIMR1281and is, therefore, partially contributing to the anti-proliferative effects of SIMR1281.
Comment 2. Figure 3g. I'm wondering why early apoptotic events (PI-/Annexin V+) are underrepresented in PI/Annexin staining after Doxorubicin treatment. Relatively low events in the right lower quadrant in treated samples and unproportionally high events for (PI+/Annexin V-) and late apoptotic (PI+/Annexin V+ ) events in control samples, that value in untreated lines usually should not exceed 2-3%.
Response: For early apoptotic events (PI-/Annexin V+), we agree that there is no detectable changes at 12h time point, but the percent of MCF7 increased from 0.9% for control cells into 4.5 % in doxorubicin-treated cells (i.e. 5 folds) and to 2.5 % in SIMR1281-treated cells (i.e. about 3 folds) after 24h of treatment. The time for detection of the early apoptotic events (extracellular efflux of phosphatidylserine) is cell-type dependent.
The high percentage of (PI+/Annexin V-) and (PI+/Annexin V+) in untreated samples could be due to the multiple washing steps that were done before adding PI and Annexin V stains (following kit's recommendation). Some damage could happen in the cell membrane, and these cells were stained with PI. Therefore, we have a control for each time point. For Doxorubicin treatment, it was reported in the literature that doxorubicin could induce necrosis, and it was suggested to happen at a low concentration of Dox. This could explain the high percentage of PI+/Annexin V- fraction after treatment with 0.06uM of Dox in our cells.
Reference: Sugimoto K, Tamayose K, Sasaki M, Hayashi K, Oshimi K. Low-dose doxorubicin-induced necrosis in Jurkat cells and its acceleration and conversion to apoptosis by antioxidants. Br J Haematol. 2002 Jul;118(1):229-38. doi: 10.1046/j.1365-2141.2002.03577.x. PMID: 12100152.
Comment 3. Figure 4b. The data is very interesting. I would suggest changing the color (to make them in different colors) of curves in the tubulin polymerization plot to clarify for readers the effect of the studied compound.
Response: Thank you for your suggestion that will improve the figure. The figure's colors were changed as suggested.